# Cannabinoids in the Inflamed Synovium Can Be a Target for the Treatment of Rheumatic Diseases

**DOI:** 10.3390/ijms25179356

**Published:** 2024-08-29

**Authors:** Livia Roseti, Giorgia Borciani, Emanuela Amore, Brunella Grigolo

**Affiliations:** RAMSES Laboratory, Rizzoli RIT-Research, Innovation & Technology Department, Istituto di Ricerca Codivilla Putti, IRCCS Istituto Ortopedico Rizzoli, Via di Barbiano, 1/10, 40136 Bologna, Italy; giorgia.borciani@ior.it (G.B.); emanuela.amore@ior.it (E.A.); brunella.grigolo@ior.it (B.G.)

**Keywords:** cannabinoids, osteoarthritis, rheumatoid arthritis, inflammation, synovium

## Abstract

The management of rheumatic diseases has noticeably changed in recent years with the development of targeted therapeutic agents, namely, biological disease-modifying antirheumatic drugs. Identifying essential signaling pathways and factors crucial for the development and progression of these diseases remains a significant challenge. Therapy could be used to delay the onset or reduce harm. The endocannabinoid system’s presence within the synovium can be identified as a suggested target for therapeutic interventions due to its role in modulating pain, inflammation, and joint metabolism. This review brings together the most pertinent information concerning the actions of the endocannabinoid system present in inflamed synovial tissue and its interaction with phytocannabinoids and synthetic cannabinoids, which can be used from a therapeutic perspective to minimize the inflammatory and pain processes typical of osteoarthritis and rheumatoid arthritis.

## 1. Introduction

Rheumatic diseases, notably, osteoarthritis (OA) and rheumatoid arthritis (RA) represent a public health issue, and the costs of prevention by reducing the risk factors or treating patients are much lower than those generated by hospitalization and surgeries [1]. Several treatments for OA and RA can alleviate disease symptoms and slow their progression. The ideal approach requires an early diagnosis, proper pharmacological treatments and non-pharmacological treatments such as exercise and physical therapy. Adequate therapy aims to obtain remission and reduce side effects [2]. However, an incomplete understanding of both diseases’ etiology makes it challenging to discover a curative treatment [3].

Considering that the significant pathological changes occurring in both OA and RA include damage to articular cartilage, subchondral bone, synovium, and articular capsule, these tissues have been identified as potential targets for pharmacological therapies.

In particular, the endocannabinoid system (ECS) in the synovium could serve as a therapeutic target due to its role in controlling pain, inflammation, and joint metabolism.

Phytocannabinoids and synthetic cannabinoids can interact with ECS components, amplifying their anti-inflammatory and analgesic effects or inhibiting pro-inflammatory and pain pathways [4]. Furthermore, cannabinoids can improve the well-being of patients by enhancing sleep, appetite, and mood [5].

However, there are still gaps regarding efficacy, the dose-dependent curve, tolerability, drug interaction, expected adverse effects, and safety [6]. Up to now, the clinical use of cannabinoids has been reported only in a few studies with small sample sizes and for short periods [7].

Only a limited number of products have been approved for therapy, and the legislation and classification process are still non-uniform and inadequate. Due to the lack of evidence-based data on safety profiles, clinicians struggle to make decisions about tailoring optimal therapy for each patient suffering from rheumatic disease [8], fearing ethical and legal issues [9].

Patients often resort to self-medication using certain products to relieve pain. It is worth noting that subjective opinions can have an impact on the reported results. Additionally, there can be issues associated with respiratory adverse effects and psychological implications, such as addiction to self-medication. Conversely, prejudice against those compounds also makes the situation complicated [8,9].

This review presents the most relevant data on the activity of endocannabinoids present in inflamed synovial tissue and their interaction with phytocannabinoids and synthetic cannabinoids from a therapeutic perspective, aiming to reduce the inflammation and pain processes typical of OA and RA.

## 2. Search Strategy and Manuscript Selection

We performed a comprehensive search according to the review strategies recommended by Gasparyan et al. [10]. We conducted the search through the PubMed, Scopus, and Web of Science databases, using the keywords “cannabinoids”, “endocannbinoids”, phytocannabinoids”, synthetic cannabinoids”, “osteoarthritis”, rheumatoid arthritis”, “rheumatic diseases”, and “synovium”. We reviewed 115 articles and included 65. Non-English sources, conference abstracts, and non-peer-reviewed sources were not included.

## 3. Rheumatic Diseases

### 3.1. Osteoarthritis

OA has been considered a disease resulting from gradual deterioration and damage (“wear and tear”), mainly impacting bone and cartilage. Recent advancements have revealed it to be a complex disorder involving the whole synovial joint [11]. Articular cartilage deterioration, subchondral bone thickening, osteophyte development, synovium inflammation, ligament damage, and capsule hypertrophy are the pathologic features of OA [12].

The etiology is multifactorial, involving genetic, constitutional, and environmental components. The widely accepted hypothesis of OA pathogenesis implicates an initial injury, frequently biomechanical, which leads to the release of mediators that activate different inflammatory pathways [13]. Evidence suggests that low-grade synovial inflammation (synovitis) plays a role in OA evolution. Common risks that pose a threat include age, gender, past joint injuries, obesity, genetic factors, malalignments, and unusual mechanical loadings.

The approaches to treat OA encompass non-pharmacological, pharmacological, and surgical treatments. Non-pharmacological options include weight loss, exercise, assistive devices, physical therapy, and acupuncture. Pharmacological interventions consist of non-steroidal anti-inflammatory drugs (NSAIDs), cyclooxygenase-2 inhibitors (COX-2 inhibitors), also known as COXIBs, glucocorticoids (GCs), opioids, chondroprotective and anti-cytokines agents. Recent therapeutic advancements encompass regenerative therapy such as platelet-rich plasma (PRP), mesenchymal stem cells (MSCs), concentrates like bone marrow concentrate (BMC) and stromal vascular fraction (SVF), and gene therapy, mainly targeting transforming growth factor beta-1 (TGF-β1). In progressive cases, surgical procedures such as arthroscopy or arthroplasty may be necessary [14,15].

### 3.2. Rheumatoid Arthritis

RA is an autoimmune disease presenting inflammation, damage, and pain in joints and other body organs and systems. The immune system’s response leads to the production of autoantibodies, including rheumatoid factors (RF) and antibodies against modified (AMPA), citrullinated (ACPA), carbamylated (aCarP), and acetylated proteins (AAPA). Highly activated monocyte/macrophage cells are present in the tissue sites involved. RA inflammation leads to joint pain and swelling, primarily in the synovium, which can cause irreversible damage to the adjacent cartilage and bone. RA is influenced by genetic, epigenetic, environmental, and stochastic factors, but its exact cause is not fully understood [16]. Treatments for RA include various drugs that help manage joint function, like disease-modifying antirheumatic drugs (DMARD). These can be conventional synthetic, biologic, or targeted synthetic. Additionally, symptom control in RA patients often involves the use of NSAIDs and GCs to reduce inflammation [16].

### 3.3. Synovial Tissue Insights into Physiological and Pathological Conditions

Synovium acts as a protective physiological barrier and plays a role in joint homeostasis [17]. This tissue encloses a lining (intima) and a sub-lining layer (subintima) composed of loose extracellular matrix (ECM) components, such as collagen, adhesin, and fibronectin. Matrix-metalloproteinase (MMP) secretion regulates tissue remodeling, increasing the degradation of collagen and non-collagen ECM components in rheumatic diseases [18]. The synovial lining layer comprises one to three strata of specialized columnar fibroblast-like synoviocytes (FLSs or “type B” synoviocytes) interspersed with lesser macrophage-like synoviocytes (MLSs or “type A” synoviocytes). FLSs regulate the ECM synthesis and synovial fluid content. This helps in lubrication and provides nutrients to the cartilage. The sub-lining layer comprises dense, loose collagens (type I and III) and adipose tissue, presenting a few cells. Microvascular blood supply, lymphatic vessels, and nerve fibers are plentiful in this layer, through which cells and nutrients pass. The synovial membrane is also a source of MSC-like cells [17]. The terminology for synovial cells is confusing as publications use terms like “type A” and “type B” synoviocytes, “FLS”, or “synovial fibroblasts”. There is also a need to clarify the term “synovium-derived MSCs”, which can be immature fibroblasts, suggesting that synovial fibroblasts can come from synovial MSCs [19].

Synovial fluid is a plasma filtrate that flows into the joint space from the sub-synovial capillaries. Its lubricating properties are attributed to the high viscosity of hyaluronic acid that synoviocytes secrete and water. Thanks to soluble molecules present within it, synovial fluid also sustains the joint. Small molecules, such as electrolytes and glucose, are present in synovial fluid at concentrations comparable to plasma. Small molecules can freely move between the synovial fluid in the joint space and the water bound to collagen and proteoglycan in cartilage. The synovial fluid’s resistance to clotting under normal conditions is due to the absence of factors in the coagulation pathway. In an inflammatory state, vasopermeability is affected, increasing the concentration of larger molecules like complement [16].

Synovitis is an inflammation that occurs continuously without any pathogens and is characterized by joint pain, swelling, limited movement, and a temperature rise. It produces many pro-inflammatory cytokines, chemokines, prostaglandins, MMPs, and pain neurotransmitters in the articular environment and promotes synovial angiogenesis and fibrosis [20].

#### 3.3.1. Synovial Tissue in OA

Synovial changes consist of abundant vascularization, villous hyperplasia, fibrosis, fibrinous exudate, cartilage and bone debris accumulation, and aggregates of lymphoplasmacellular infiltrates [12]. The activation and proliferation of FLSs leads to the release of pro-inflammatory cytokines, chemokines, nerve growth factor (NGF), and proteolytic enzymes. This results in the spread of inflammation, disturbance of the synovial microenvironment, and damage to the cartilage ECM [21]. Macrophages are recruited along with T cells, with fewer mast, B, and endothelial cells. Macrophages polarize into M1 or M2 subtypes, with an imbalanced ratio between M1 and M2. The accumulation of M1 leads to an increase in inflammation by releasing pro-inflammatory cytokines and activating cells in the local environment [22]. Synovial fibrosis develops in the late stages of OA. The process involves the restructuring of the tissue architecture by osteophyte formation, pathological type I collagen deposition, and fibroblast proliferation, resulting in hyperplasia, stiffness, and limited function [18,23].

Synovial fluid viscosity diminishes due to decreased lubricin and hyaluronic acid concentration, which also changes in size [24].

#### 3.3.2. Synovial Tissue in RA

Synovial tissue expands due to the increase in and activation of both MLSs and FLSs, which produce a variety of pro-inflammatory cytokines, MMPs, and other mediators inducing joint destruction. Adaptive immune cells invade the synovial sub-lining, forming the characteristic “pannus” at the cartilage–bone interface [25]. FLSs and activated immune cells may form an ectopic lymphoid structure (ELS) in the synovial tissue. B and plasma cells are also present in the synovium, producing antibodies like RF, and ACPAs, which help form immune complexes and trigger the complement cascade [26].

Macrophages and T-cells are often unbalanced. M1/Th1 activation results from the interaction of Toll-like receptor and interferon (IFN) signaling in an inflammatory environment, leading to the overproduction of pro-inflammatory cytokines. On the other hand, M2/Th2 activation leads to the secretion of anti-inflammatory molecules, ultimately leading to the clinical remission of RA [27].

Joint swelling and capsule stretching caused by synovial fluid overproduction can result in pain. FLSs facilitate the flow of electrolytes and water through sodium–potassium–chloride cotransporter1 (NKCC1) and aquaporin-1, which in turn leads to cell swelling and cytotoxic edema [28].

## 4. Cannabinoids

Cannabinoids are mediators that, according to their origin, exist in three types: endogenous (or endocannabinoids) produced by the human body, exogenous (or phytocannabinoids) naturally present in the *Cannabis sativa* plant, and synthetic cannabinoids designed and synthesized to preserve some properties of phytocannabinoids without the typical psychoactive effects [29] (Figure 1).

### 4.1. Endocannabinoids

Endocannabinoids are highly lipophilic lipid messengers that act by binding and activating specific membrane receptors and whose metabolism and transport are regulated by specific enzymes. The storage of endocannabinoids is different from that of conventional neurotransmitters, as they are synthesized locally on demand in the damaged sites [29]. Cannabinoids, their receptors, and metabolite enzymes constitute ECS, which drives many physiological processes, including the response to injury or disease and the modulation of inflammation, pain, and the immune system [30].

The two most studied endocannabinoids are N-arachidonoylethanolamide (anandamide, AEA) and 2-arachidonoylglycerol (2-AG), which damaged tissues produce in response to nociceptive signals [9]. FAAH is the primary enzyme responsible for terminating the biological actions of AEA, whereas 2-AG is the enzyme responsible for terminating 2-AG. When AEA and 2-AG are enzymatically degraded, they release arachidonic acid, ethanolamine, and glycerol, respectively [31]. Other biochemically similar endocannabinoids are 2-AG ether, virodhamine, and N-arachidonoyl dopamine [32].

The anti-nociceptive, -inflammatory, and immunomodulatory effects of cannabinoids are correlated to their capacity to move binding ligands away from the endocannabinoid receptor, hindering their signaling [33]. The most abundant and distributed receptors are the cannabinoid (CB)1 and CB2 receptors (CB1R and CB2R), belonging to the family of transmembrane receptors (7TM) coupled to the G protein (GPCR) [34]. CBRs are mainly distributed in central nervous system cells, modulating the release of excitatory or inhibitory neurotransmitters [30]. They are also in smaller quantities in some peripheral organs and tissues [35,36]. CB2Rs are principally expressed at the peripheral level, particularly in immunocompetent cells, controlling the release of inflammatory cytokines. Ligands can interact with receptors in different ways. Agonists can have varying degrees of efficacy, activating the receptor, whereas antagonists have no efficacy and prevent activation. When an agonist binds to the receptor, it causes a modification that can trigger enzymes or open ion channels nearby. The agonist and receptor bond are reversible, making it weak. Antagonists, on the other hand, prevent receptor activation. Receptors can also be active without an activating ligand, showing “constitutive” activity, which inverse agonists can reduce. Inverse agonists produce the opposite effect of an agonist. Additionally, partial agonists can bind to a target without fully activating the receptor [29]. The responsibility of preserving basal endocannabinoid signaling falls on AEA, which binds to the CB1R and CB2R. In high amounts, AEA can act as a complete agonist for the transient receptor potential vanilloid 1 (TRPV1), which functions by modulating ion passage through its channel [37]. Peroxisome proliferator-activated receptors alpha and gamma (PPARa and PPARg), G-protein-coupled receptor 55 (GPR55), and G-protein-coupled receptor 18 (GPR18) are among other receptors. 2-AG, unlike AEA, is capable of fully agonizing CB1R and CB2R [38,39].

### 4.2. Phytocannabinoids

Phytocannabinoids are lipophilic molecules with a terpenophenolic structure; they generally mimic the effects of endocannabinoids by binding to the same receptors. The *Cannabis sativa* plant produces over 100 cannabinoids like Δ9-tetrahydrocannabinol (Δ9-THC, THC), cannabidiol (CBD), cannabigerol (CBG), cannabinol (CBN), and cannabichromene (CBC). THC and CBD are the two cannabinoids investigated for their health benefits [40].

THC is the psychoactive cannabinoid of the *cannabis* plant, which acts as an agonist on both CB1R and CB2R. The concentration of THC contained in the plant is generally used as a measure of the potency of *cannabis* and, consequently, of its effects. THC can be responsible for adverse psychoactive events, depending on the dose and the patient’s previous tolerance [40].

CBD has a low affinity for classical CB1R and CB2R but can bind to transient receptor potential (TRP) channels: it has agonistic effects on TRP vanilloid-4 (TRPV1-4) and -1 (TRPV1-1) and antagonistic effects on the TRP melastatin-8 (TRPM-8) [41,42].

CBG acts as a partial agonist and acts via the CB2R in macrophages or “antagonizes” the effects of endogenous or synthetic cannabinoids [43].

THC has a lesser affinity for the CB1R than CBN but is more effective at inhibiting the activity of adenylyl cyclase. Depending on the study and the tissue, CB2R activity seems to have a changeable profile. CBN acts as an agonist for many TRPV channels, stimulating the Ca^2+^ influx and activating Ca^2+^-dependent pathways. CBN can also stimulate the TRPA1 channels [44].

The efficacy of CBC, which is a selective CB2R agonist, is superior to that of THC. CBC could induce the CB2R-mediated modulation of inflammation, contributing to the therapeutic effects of some *cannabis* preparations [45].

#### Release Methods and Formulations

Phytocannabinoids display some features that may restrict their utilization from a clinical perspective. Highly lipophilic molecules are poorly soluble in water, highly unstable, and susceptible to degradation by light, temperature, or autooxidation [5,46]. Moreover, they show poor systemic bioavailability. Association with other compounds and formulation types (emulsification, micellization, encapsulation in liposomes or nanoparticles) can increase solubility and stability, improve pharmacokinetic profile and bioavailability, and enhance targeted delivery. The use of nanotechnology in drug delivery is exploding, with nano-formulation systems that incorporate polymeric and lipid-based nanoparticles, ethosomes, and cyclodextrins [47].

Commonly used administration forms are inhalation (smoke or vaporization) and oral administration. Gold-mucosal, transdermal topical, and rectal administration are less common than the previous ones [46]. The pharmacokinetics and pharmacodynamics of the drug change depending on the type of administration. Oral or topical administrations of a CBD-based oil in arthritis animal models showed reduced symptoms and pain. Transdermal administration ensured higher absorption than oral administration in the same animal model [5].

### 4.3. Synthetic Cannabinoids

This literature review does not cover synthetic psychoactive substances that can cause multiple system complications that are commonly found in abusers [48].

Research on the potential mechanism of action of cannabinoids has sparked interest in finding synthetic molecules that can replicate the effects of cannabinoids. The aim was to create and produce molecules that could maintain the therapeutic properties of cannabinoids without causing unwanted effects such as addiction. The first attempts involved synthesizing molecules like phytocannabinoids, including nabilone and HU-210 (an analog of THC) [29]. Subsequently, only the specific sites interacting with receptors were replicated, as seen with WIN-55212-2. Several derivatives with a structure like WIN-55212-2 have been created and tested, including JWH-073, JWH-018, and JWH019 [49]. CB2R agonist JWH133 was effective in reducing inflammation [50].

Despite the most recent generation of synthetic cannabinoids, more investigation is needed to determine their long-term impact, as well as their pharmacological and toxicological profiles [51].

## 5. Anti-Inflammatory and Antalgic Effects of Cannabinoids in the OA and RA Synovium

AEA and 2-AG were found in the synovial fluid of two OA or RA patient populations but not in healthy patients [36]. These endocannabinoids showed increased levels in OA animals’ synovial membranes or fluids [52]. A comparable situation occurred in vitro for AEA’s synthesis and degradation enzymes, like FAAH and COX-2 [53]. AEA has a short half-life due to enzymatic degradation. Therefore, targeting its degrading enzymes is a promising strategy. For example, URB597 (3′-(aminocarbonyl) [1,1′-biphenyl]-3-yl)-cyclohexylcarbamate) inhibitor markedly reduced FAAH activity in the synovial membrane of OA and RA patients [36]. AEA’s activity was enhanced in primary synoviocytes when FAAH was inhibited, but not in FLSs. In a COX-2-dependent pathway, TRPV1 and TRPA1 receptors can mediate AEA consequences [54]. In tissue inflammation, COX-2 plays a significant role in producing prostanoid from arachidonic acid to prostaglandin H, which then gets converted into prostaglandin E2 [55]. Compared to controls, the COX-2 presence is higher, localizing predominantly in the FLSs of patients with OA and RA. Elevation of AEA levels and de-sensitization of the TRP channel may prevent both FAAH and COX-2 from producing pro-inflammatory mediators by inhibiting them [54].

The synovial membrane of OA or RA patients contained CB1R and CB2R, not synovial fluid. This suggests that these receptors are synthesized on demand from membrane-bound phospholipids rather than stored in the cells. Their release was vesicle-independent [36,56]. Stimulating both receptors in tumor necrosis factor-alpha (TNF-α)-induced FLSs led to activating specific signaling pathways that induce IL-8 protein. Extracellular signal-regulated kinase (ERK)-1,-2, and JNK MAPK’s phosphorylation was the initiating factor for this process. As a piece of evidence, the non-selective agonist HU210 phosphorylated ERK1 and ERK2 in FLSs in a PTX-dependent manner via the CB1Rs; in addition, the CB1 antagonist SR141716A significantly blocked this activity [21,53]. Activating the CB2R with HU-308 agonist can prevent the proliferation of OA FLSs and their secretion of IL-6 and MMPs [57]. FLSs of patients with RA showed a higher expression level of the CB2Rcompared to OA [57]. JWH-015, a CBR agonist, was shown to inhibit pro-inflammatory activity in interleukin-1 (IL-1β)-induced mouse RA FLS, partly through the GC receptor [57]. JWH133, another CB2R agonist, inhibited TNF-α-stimulated RA FLSs from releasing inflammatory mediators. These findings suggested that inflammation could stimulate AR FLSs to express more CB2R as a feedback mechanism to counteract synovitis and joint destruction [58]. CB2R knockdown in human RA FLSs resulted in reduced pro-inflammatory factors and degrading enzymes [58]. In synovium, nerve fibers express CB2Rs, as evidenced by in vitro studies [59].

CB1R, CB2R, TRPV1, GPR55, and PPARa were found in FLSs of the dorsal synovium of the equine metacarpophalangeal joint, which can develop OA due carrying heavy loads during movement. CB1R was present in some of the horses investigated, but not all [60]. An *RPV1* gene variant has been linked to the risk of developing OA [60], according to a study [61].

The non-selective synthetic cannabinoids (CP55,940 and WIN55,212-2) can exert significant anti-inflammatory effects on cultured IL-1β RA FLS by inhibiting IL-6 and IL-8 via a non-CB1R/CB2R-mediated mechanism [62].

Mice with collagen-induced arthritis (CIA) were utilized as an RA model to assess CBD’s effects [63]. When CBD is administered intraperitoneally or orally after the onset of clinical symptoms, it can block arthritis development dose-dependently. Cultured knee synoviocytes from CBD-treated mice showed a reduced release of TNF compared to untreated controls [64].

CBD demonstrated the capacity to raise calcium levels and decrease the production of inflammation mediators in human RA FLS cultures via activation of TRPA1 and mitochondrial targets. Pre-treatment with TNF could increase these effects [65]. Moreover, CBD induces regulatory T cells, which have a role in regulating or suppressing other cells in the immune system [66].

The synthetic THC-derivative, known as dimethylheptyl-THC-11-oic acid (DMH-11C) or ajulemic acid, can prevent inflammation and is not psychoactive. It has been shown to decrease pannus formation in mouse models of acute (cytokine-induced) and chronic (adjuvant-induced) inflammation [67].

## 6. ECS Effects on Immune Cells in RA

CB2R, more predominant in immune cells in RA, is a promising target treatment for reducing immune cell activity and cytokine production. Current treatments for RA, including GCs, often disrupt cytokine production or signaling, leading to infection- or immune-related side effects. Therefore, a treatment that reduces these risks would be preferable [66]. CB2R is present in B cells during RA, influencing plasma cell number, homing, and antibody amount [57]. CB2R can also reduce the activation of NOD-, LRR-and pyrin-domain-containing protein 3 (NLRP3 inflammasome), a central component of the innate immune system, leading to subsequent IL-1β release, decrease TNF or lipopolysaccharide (LPS) cell response, and enhance the clearance of apoptotic cells [66]. In T cells, CB2 inhibits B cell proliferation, differentiation, and nuclear factor kappa-light-chain-enhancer of activated B cells (NF-kB) that mediates innate and adaptive immunity by initiating an inflammatory response to proinflammatory signals. CB2R activates migration, cell energy supply, and response to T-cell-independent antigens in B cells. CB2R agonists induce beneficial anti-inflammatory effects by downregulating proinflammatory cytokine production and reducing mobilization and migration of immune cells to inflammation sites [66,68].

A common dinucleotide polymorphism, Q63R, in the CB2R gene (*CNR2*) has been shown to alter the ability of CB2R to exert its inhibitory function on T lymphocytes. A study investigated the association between Q63R and RA in the Lebanese population, suggesting a role of this gene polymorphism in the etiology of RA, thus supporting its potential use as a pharmacological target for selective agonists in clinical practice [69].

In contrast to CB2R, CB1R induces proinflammatory effects by promoting macrophage M1 polarization, generating reactive oxygen species, enhancing the fibrotic process, and TLR4 signaling, a primary activator of the innate immune response. However, it reduces pain and depression. CB1R agonists promote proinflammatory signaling that CBD can prevent, enhancing their efficacy. CB1 antagonists exert anti-inflammatory effects by enhancing β_2_-adrenergic signaling secondary lymphoid organs. However, they provoke psychiatric issues. Therefore, peripherally restricted CB1 antagonists are under investigation. In addition, CBD demonstrated anti-arthritic effects and the ability to activate noncannabinoid pathways [66].

Based on that evidence, Lowin et al. suggested using a combination of a CB2 agonist to reduce cytokine production, a peripheral CB1 antagonist to block harmful CB1 signaling and bolster the anti-inflammatory effects of CB2 through the activation of β_2_-adrenergic receptors and CBD to elicit cannabinoid receptor-independent anti-inflammatory effects [66].

## 7. Animal Models

ECS is universal to all animal species except insects. Our understanding of animals’ ECS has not only deepened our knowledge but also paved the way for innovative approaches that could revolutionize pain management and the treatment of inflammatory diseases.

Although the CB1R structure remains consistent across all mammalian species, the protein sequences of the CB2RS vary significantly in humans, rats, and dogs. Furthermore, the CB2-Rs in humans, mice, and rats differ in peptides, mRNA sizes, gene structure, and pharmacology [70,71].

The complete gene structure of human CB2R (*CNR2*) has yet to be fully characterized and is still unavailable for some animal species. Human CNR2 gene structure and transcription mechanisms differ from rodent *CNR2* genes. The differential promoter activities and non-homologous 5′UTR and 3′UTR sequences in humans and rodents might explain the differential anti-nociceptive effects of CB2 agonists in human and rodent models [70].

A meta-analysis showed significant differences in ligand-binding affinities between human and rat CB2 [72].

These interspecies differences could compromise the translation of ECS studies to human participants, as humans may exhibit fewer effects than animals [71].

## 8. Patient’s Expectation

The available approaches for OA and RA treatment open the critical question of how to manage the different therapeutic strategies in personal situations. Patients’ preferences should be respected to enhance therapy adherence, addressing their physical, emotional, social, and spiritual well-being. Moreover, OA and RA are heterogeneous diseases characterized by variable clinical features, biochemical/genetic characteristics, and treatment responses.

All these aspects underline that patients are more likely to choose a tailored approach than a ‘one size fits all’ approach based on these aspects. Clinicians should consider prescribing both pharmacological and non-pharmacological interventions, considering important aspects such as comorbidity, combination therapy, and contraindications in patients who may be unwilling to take additional drugs.

## 9. Clinical Implications

Clinicians need to assess the potential risks and benefits of using cannabinoids for OA and RA treatment. Essential criteria like age, severity of disease, failure of standard medical therapy, and concurrent psychiatric disorders should be considered when selecting patients. Finally, clinicians should evaluate patients’ expectations before and after this treatment.

## 10. Conclusions

This review focused on synovial tissue ECS and its interaction with phytocannabinoids and synthetic cannabinoids, which are potential therapeutic compounds for treating OA and RA. Some evidence suggests that the ECS has the potential to help address certain rheumatic diseases because of its several targets for treatment in inflamed joints.

However, preclinical and clinical studies are needed to develop CBS-based drugs and overcome ethical, political, and legislative issues related to their use.

## Figures and Tables

**Figure 1 ijms-25-09356-f001:**
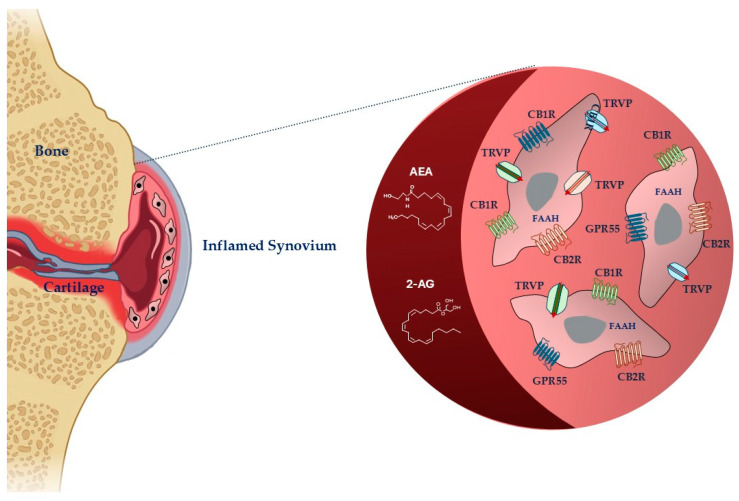
Schematic representation of cannabinoid system in fibroblast-like synoviocytes in an inflamed synovium. Legend: AEA: anandamide; 2-AG: 2-arachidonoylglycerol, FAAH: fatty acid amide hydrolase; CB1R: cannabinoid receptor-1; CBR2: cannabinoid receptor-2; TRPV: transient receptor potential vanilloid; GPR55: G-protein-coupled receptor 55. The image was created with BioRender.com.

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
