# Peer review of "Cannabinoids in the Inflamed Synovium Can Be a Target for the Treatment of Rheumatic Diseases"

_ijms, 2024, doi:10.3390/ijms25179356_

Round 1

Reviewer 1 Report

Comments and Suggestions for Authors

The review describes the anti-inflammatory activities of cannabinoids in the synovial milieu.

The paper is clear and well structured.

In section 4, I would recommend including observations on the anti-inflammatory properties of Ajulemic acid (1) in adjuvant-induced arthritis in murine models. Similarly, It has been  described that non-selective synthetic cannabinoids (CP55,940 and WIN55,212-2) exert a significant anti-inflammatory effect on cultured synoviocytes by inhibiting IL-6 and IL-8 (2). However, this effect does not seem to be mediated by the known cannabinoid receptor systems, such as CB1, CB2, PPAR-gamma, or TRPV1.

1)      Zurier RB, Rossetti RG, Lane JH, Goldberg JM, Hunter SA, Burstein SH. Dimethylheptyl-THC-11 oic acid: a nonpsychoactive antiinflammatory agent with a cannabinoid template structure. Arthritis Rheum. 1998 Jan;41(1):163-70.

2)      Selvi E, Lorenzini S, Garcia-Gonzalez E, Maggio R, Lazzerini PE, Capecchi PL, Balistreri E, Spreafico A, Niccolini S, Pompella G, Natale MR, Guideri F, Laghi Pasini F, Galeazzi M, Marcolongo R. Inhibitory effect of synthetic cannabinoids on cytokine production in rheumatoid fibroblast-like synoviocytes. Clin Exp Rheumatol. 2008 Jul-Aug;26(4):574-81.

Author Response

Dear Reviewer,

Thank you for the revision and for your kind suggestions to improve the quality of our manuscript. We have tried our best to follow all comments and recommendations.

In the following, we have italicized your comment and inserted our answer in regular font.

1.In section 4, I would recommend including observations on the anti-inflammatory properties of Ajulemic acid (1) in adjuvant-induced arthritis in murine models. Similarly, It has been described that non-selective synthetic cannabinoids (CP55,940 and WIN55,212-2) exert a significant anti-inflammatory effect on cultured synoviocytes by inhibiting IL-6 and IL-8 (2). However, this effect does not seem to be mediated by the known cannabinoid receptor systems, such as CB1, CB2, PPAR-gamma, or TRPV1.

We thank the Reviewer for his/her positive feedback and comments. We included some new sentences considering the suggestions and added the recommended references.

Reviewer 2 Report

Comments and Suggestions for Authors

The manuscript “Cannabinoids in the inflamed synovium can be a target for the treatment of rheumatic diseases” by Livia Roseti and colleagues is a review on the use of phytocannabinoids and synthetic cannabinoids to reduce pain and inflammation in osteoarthritis and rheumatoid arthritis. This review is very interesting because it discusses the use of molecules that are at the interface between immune and nervous system and that are not yet extensively explored for therapy. This is a very important topic both in debilitating autoimmune disorders and in cancer.

I think the manuscript would improve is revised by an English mother tongue person.

Authors refer the use of animal models to address the role of cannabinoid treatment on RA, I would suggest to point out a critical issue of these studies related to the differences between species, for example for what concerns receptor expression.

Worth to discuss possible roles of the different cannabinoid receptor isoforms and their expression on lymphocytes, and how cannabinoid derived therapy can impact on tertiary lymphoid structure generation/persistence in the synovia, and thus modulate acquired immune response and autoantibody secretion.

Because authors refer genetic factors related to RA, I was wondering if there are cannabinoid receptor variants, already described, interfering with cannabinoid therapy efficacy.

Minor comments:

line33: “In particular, the ECS in the synovium”- give the complete name of the abbreviation

line 52: “The situation is then complicated by people using non-medical cannabis and other substances that can generate prejudice” which other drugs can have their effect altered by combining cannabinoids? Please give references for this observation

line 169: “Cannabis sativa” consider to write in italic form as normally used for the specie identification

Comments on the Quality of English Language

I think the manuscript would improve is revised by an English mother tongue person.

Author Response

Dear Reviewer,

Thank you for the revision and for your kind suggestions to improve the quality of our manuscript. We have tried our best to follow all comments and recommendations. Please consider that other Reviewers have required some variations.

We hope that the changes meet with approval. In the following, we have italicized all your comments and inserted our answers in regular font.

1. “I think the manuscript would improve is revised by an English mother tongue person

We thank the reviewer for his comment and have had the manuscript checked by a native-speaker translator from our institute.

2. “Authors refer the use of animal models to address the role of cannabinoid treatment on RA, I would suggest to point out a critical issue of these studies related to the differences between species, for example for what concerns receptor expression.

We thank the Reviewer for bringing this point to our attention. We included a paragraph in the text (“Animal Models”) that discusses the issues related to the use of animal models in this field.

3. “Worth to discuss possible roles of the different cannabinoid receptor isoforms and their expression on lymphocytes, and how cannabinoid derived therapy can impact on tertiary lymphoid structure generation/persistence in the synovia, and thus modulate acquired immune response and autoantibody secretion.

We thank the Reviewer for this comment, which aims to improve the quality of the review. We added a new paragraph, “ECS effects on immune cells in RA,” discussing cannabinoid receptor isoforms and their expression on lymphocytes and how cannabinoid-derived therapy can impact lymphoid structures and thus modulate acquired immune response and autoantibody secretion.

4. “Because authors refer genetic factors related to RA, I was wondering if there are cannabinoid receptor variants, already described, interfering with cannabinoid therapy efficacy.

We thank the reviewer for the interesting question. We found a reference related to this point and added it to the manuscript.

“Ismail M, Khawaja G. Study of cannabinoid receptor 2 Q63R gene polymorphism in Lebanese patients with rheumatoid arthritis. Clin Rheumatol. 2018 Nov;37(11):2933-2938. doi: 10.1007/s10067-018-4217-9. Epub 2018 Jul 21. PMID: 30032418”.

5. “line33: “In particular, the ECS in the synovium”- give the complete name of the abbreviation

We thank the Reviewer for pointing out the error. We explained the acronym and verified all the other ones in the text.

6. “line 52: “The situation is then complicated by people using non-medical cannabis and other substances that can generate prejudice” which other drugs can have their effect altered by combining cannabinoids? Please give references for this observation

We are grateful to the Reviewer for raising this point with us. Since our sentence refers to the non-medical use of Cannabis, we deleted the term "medical" to avoid confusion and rearranged the text.

7. line 169: “Cannabis sativa” consider to write in italic form as normally used for the specie identification

We thank the Reviewer for pointing out the error. We changed the species identification to italics.

Reviewer 3 Report

Comments and Suggestions for Authors

This review summarises the different aspects of the ECS in OA and RA. The manuscript is well structured and written, but needs some canges and additions to improve the quality.

Line 32: ECS acronym is not explained. All abbreviations need to be revisited.

Lines 48-49: is medical cannabis used in rheumatic diseases-associated pain? If yes, please provide references. If not, this needs to be removed as non-relevant.

Fig1 and its legend provide very limited information. The authors need to either explain the role of these molecules in the legend or place the figure after section 3.2. A it stands, it is very difficult for the reader to follow.

The authors describe the physiological role of ECS system in section 3. However, they include general information which is unclear if it is focused on OA and RA. It would be more useful if the alterations of ESC when OA/RA occur could be explained in depth.

What is the role of ECS and the relative formulations in other types of cell that are implicated in OA/RA? A section for this would also add to this work.

A summative table which will include the effects of the different ECS-related molecules and formulations on human/animal cells, animal models and human trials is essential.

Author Response

Dear Reviewer,

Thank you for the revision and for your kind suggestions to improve the quality of our manuscript. We have tried our best to follow all comments and recommendations. Please consider that another Reviewer has required other variations, so we have also modified the text at points you have not mentioned. We hope that the changes meet with approval. In the following, we have italicized all your comments and inserted our answers in regular font.

1. “Line 32: ECS acronym is not explained. All abbreviations need to be revisited.

We thank the Reviewer for outlining the mistakes. We explained the acronym and checked all the others in the text.

2. “Lines 48-49: is medical cannabis used in rheumatic diseases-associated pain? If yes, please provide references. If not, this needs to be removed as non-relevant.”

 We thank the Reviewer for bringing this point to our attention. Since our sentence refers to the non-medical use of Cannabis, we deleted the term "medical" to avoid confusion and rearranged the text.

3. “Fig1 and its legend provide very limited information. The authors need to either explain the role of these molecules in the legend or place the figure after section 3.2. A it stands, it is very difficult for the reader to follow.”

 We thank the Reviewer for this comment and moved the Figure after Section 3.2.

4. “The authors describe the physiological role of ECS system in section 3. However, they include general information which is unclear if it is focused on OA and RA. It would be more useful if the alterations of ESC when OA/RA occur could be explained in depth “

We thank the Reviewer for this comment. Section 3 is a general section describing the physiological role of ECS. The following section (4) is instead dedicated to ESC in the synovium of the two pathological conditions. Since our review mainly focuses on highlighting the role of ECS in the synovium, we pointed out the alterations of ECS occurring in this tissue in OA/RA.

5. “What is the role of ECS and the relative formulations in other types of cells that are implicated in OA/RA? A section for this would also add to this work. “

We thank the Reviewer for this comment. As reported above, our review mainly aims to highlight the role of ECS in synovial tissue. For this reason, we did not include details about other cell types implicated in the two pathologies except brief mentions. However, following your suggestion, we will soon consider writing a manuscript on the role of ECS and the relative formulations in other cell types.

6. “A summative table which will include the effects of the different ECS-related molecules and formulations on human/animal cells, animal models and human trials is essential.”

We thank the Reviewer for the suggestion. However, data on the effectiveness of ECS and related molecules and formulations in human/animal cells, animal models, and human trials in the synovium of OA and RA patients are limited. Therefore, we opted to use a narrative description that tries to highlight the scientific evidence on each molecule in a common thread.

Reviewer 4 Report

Comments and Suggestions for Authors

The authors have developed a well-written review to provide a comprehensive understanding of how cannabinoids can be utilized in the treatment of rheumatic diseases, focusing on both natural and synthetic options

However, I would like to make a few observations before recommending their work for publication.

1. Since it is presented as a review paper, your manuscript should be structured according to international scientific standards: Introduction, Methods, Results, Discussion, and Conclusions.

2. Could the authors provide details of the keywords used, as well as the databases consulted to carry out the review?

3. Could the authors provide a PRISMA flowchart?

4. I ask the authors to detail the selection process during the search for articles: Total number of articles found, articles selected by reading the title and abstract once duplicates are excluded, and articles finally included in the review once the complete reading of the articles has been carried out.

5. Have the authors thought about Discussing patients' expectations? The scientific evidence shows us the importance of collecting this information, and I advise the authors to discuss this in the corresponding section. I leave you with one useful work to draw upon: doi.org/10.47197/retos.v46.93950 

6. Could you add a section on "Clinical Implications"?

7. I believe that other conservative treatments, which could be complementary, should be discussed and/or briefly presented. Here are some references of  novel papers on Fibromyalgia , back pain, and chronic musculoskeletal pain: DOI:10.1016/j.msksp.2024.103160 ; DOI: 10.1097/TGR.0000000000000435 ; DOI: 10.3390/jcm12206478 ; DOI: 10.1097/PHM.0000000000002239

8. Reference 59 must be spelled correctly in the bibliography section.

Comments on the Quality of English Language

No comments

Author Response

Dear Reviewer,

Thank you for the revision and for your kind suggestions to improve the quality of our manuscript. We have tried our best to follow all comments and recommendations. Please consider that another Reviewer has required other variations, so we have also modified the text at points you have not mentioned. We hope that the changes meet with approval. In the following, we have italicized all your comments and inserted our answers in regular font.

1. “Since it is presented as a review paper, your manuscript should be structured according to international scientific standards: Introduction, Methods, Results, Discussion, and Conclusions”.

We thank the Reviewer for his/her kind comment. However, the manuscript has been structured strictly according to the journal's instructions to authors.  

2. “Could the authors provide details of the keywords used, as well as the databases consulted to carry out the review?”

We thank the Reviewer for the suggestion. We added a new paragraph (2) providing details on the keywords and the database used to carry out the review.

3. “Could the authors provide a PRISMA flowchart?”

Thank you for the suggestion. However, as for Instructions to Authors of the Journal, the PRISMA flowchart is indicated only for scoping review and is unsuitable for the present one.

4. “I ask the authors to detail the selection process during the search for articles: Total number of articles found, articles selected by reading the title and abstract once duplicates are excluded, and articles finally included in the review once the complete reading of the articles has been carried out.”

We thank the Reviewer for the comment. As reported above (n.2), we added a new paragraph (2. Search strategy and manuscript selection) that provided details on the keywords and the database used to carry out the review. We also added a suitable reference.

5. “Have the authors thought about Discussing patients' expectations? The scientific evidence shows us the importance of collecting this information, and I advise the authors to discuss this in the corresponding section. I leave you with one useful work to draw upon: doi.org/10.47197/retos.v46.93950”.

We thank the Reviewer for the suggestion. We added a section on patient's expectations.

6. “Could you add a section on "Clinical Implications?"

 We thank the Reviewer for the suggestion. We added a section on clinical implications.

7. “I believe that other conservative treatments, which could be complementary, should be discussed and/or briefly presented. Here are some references of novel papers on Fibromyalgia , back pain, and chronic musculoskeletal pain: DOI:10.1016/j.msksp.2024.103160; DOI: 10.1097/TGR.0000000000000435; DOI:10.3390/jcm12206478; DOI: 10.1097/PHM.0000000000002239

We thank the Reviewer for the comment. We agree with the importance of conservative treatments for rheumatic disease pain, which could be complementary. However, due to the topic of the review, which mainly focuses on ECS in OA and RA synovium, we have considered limiting this aspect in the Introduction section by adding one of your suggested and most related references.

8. “Reference 59 must be spelled correctly in the bibliography section”.

We thank the Reviewer for pointing out the error in the reference, and we corrected it accordingly.

We thank the Reviewer for helping us improve the quality of the manuscript. We hope this new version is more interesting for readers and suitable for publication.

Round 2

Reviewer 4 Report

Comments and Suggestions for Authors

The authors have improved with their current version the previous version of their manuscript, so I recommend its publication.

Congratulations.

Comments on the Quality of English Language

No comments

Author Response

We thank the Reviewer for recommending the publication of our manuscript